# Developing digital contact tracing tailored to haulage in East Africa to support COVID-19 surveillance: a protocol

Adrian Muwonge ,[1,2] Christine Mbabazi Mpyangu,[3] Allen Nsangi ,[4,5] Ibrahim Mugerwa,[6] Barend M deC Bronsvoort,[7] Thibaud Porphyre,[8] Emmanuel Robert Ssebaggala,[9] Aggelos Kiayias,[2] Erisa Sabakaki Mwaka,[10] Moses Joloba[11]

For numbered affiliations see end of article.

**Correspondence to**
Adrian Muwonge;
adrian.muwonge@roslin.ed.ac.uk

## ABSTRACT

**Introduction** At the peak of Uganda's first wave of SARS-CoV-2 in May 2020, one in three COVID-19 cases was linked to the haulage sector. This triggered a mandatory requirement for a negative PCR test result at all ports of entry and exit, resulting in significant delays as haulage drivers had to wait for 24–48 hours for results, which severely crippled the regional supply chain.

To support public health and economic recovery, we aim to develop and test a mobile phone-based digital contact tracing (DCT) tool that both augments conventional contact tracing and also increases its speed and efficiency.

**Methods and analysis** To test the DCT tool, we will use a stratified sample of haulage driver journeys, stratified by route type (regional and local journeys).

We will include at least 65% of the haulage driver journeys ~83 200 on the network through Uganda. This allows us to capture variations in user demographics and socioeconomic characteristics that could influence the use and adoption of the DCT tool. The developed DCT tool will include a mobile application and web interface to collate and intelligently process data, whose output will support decision-making, resource allocation and feed mathematical models that predict epidemic waves.

The main expected result will be an open source-tested DCT tool tailored to haulage use in developing countries. This study will inform the safe deployment of DCT technologies needed for combatting pandemics in low-income countries.

**Ethics and dissemination** This work has received ethics approval from the School of Public Health Higher Degrees, Research and Ethics Committee at Makerere University and The Uganda National Council for Science and Technology. This work will be disseminated through peer-reviewed publications, our websites https://project-thea.org/ and Github for the open source code https://github.com/project-thea/.

---

## BACKGROUND

The COVID-19, caused by the SARS-CoV-2 virus, is a global pandemic that has stretched public health systems worldwide, often

## STRENGTHS AND LIMITATIONS OF THIS STUDY

⇒ This is the first study that fully documents the development of a digital contact tracing (DCT) tool in consultation with stakeholders on the African continent.

⇒ It brings together a multidisciplinary team to provide a solution to an old and neglected problem 'the role of haulage in transmission of infectious diseases'. Therefore, although the proposed DCT tool is linked to the COVID-19 pandemic, its utility goes far beyond the current pandemic.

⇒ The ever-changing epidemiological landscape of COVID-19 means that the study design is subject to adjustments to accommodate the prevailing public health demands. These will however be openly highlighted and discussed in the resulting publications.

beyond carrying capacity.[1 2] The transmission rates and rapid emergence of novel strains are particularly challenging all conventional epidemiological wisdom and, by extension, effective deployment of routine contact tracing tools.[3] At the peak of Uganda's first COVID-19 wave, community transmission was significantly attributed to haulage drivers' interaction and seeding the infection into communities along the road network, linked to service stations.[4] This triggered the requirement to demonstrate proof of health; a negative COVID-19 PCR test result for all haulage drivers at Uganda's ports of entry and exit (POEs). This inevitably resulted in a pile-up that brought the entire regional supply chain to a halt. For example, in May 2020, Uganda had a queue of up to 47 km of haulage trucks at Busia and Malabaorder ports between Uganda and Kenya,[5] which prompted the COVID-19 task force at the Ministry of Health, in consultation with the National Logistic Platform (de facto trade union for haulage drivers) to develop a mitigation plan

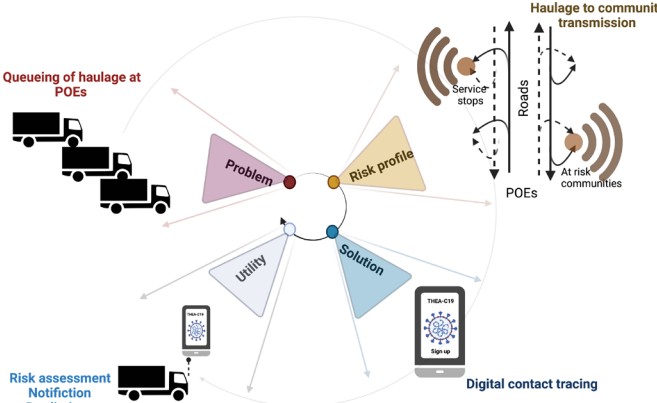

**Figure 1** The study is responding to a public health and economic challenge linked to haulage sector as shown above. This group represents a risk of introducing COVID-19 and its variants to communities along the road infrastructure such as service stations, resting points and restaurants. This risk is exacerbated by delays which cause queuing up at POEs. Our solution is aimed at improving the efficiency of conventional contact tracing using digital contact tracing. The utility- is that this would allow for timely and robust risk assessment, automatic notification of primary and secondary contacts of identified cases as well as prediction of hotspots along the haulage network to support public health preparedness. The image was generated using Biorender https://biorender.com.

to the problem as contextualised in (figure 1). This plan aimed to limit the risk of virus and novel variant introduction into communities along the road network . Haulage drivers were allowed to obtain a single negative test result valid for 14 days. The success of this plan had to be underwritten by a faster contact tracing approach for cases and their immediate contacts to support decision-making and resource allocation[6] (figure 1).

Effective and timely contact tracing slows the progression of community spread which addresses the economic and public health concerns that arise from mandatory lockdowns.[7] In this regard, case investigation and faster contact tracing are considered primary mitigating interventions to control and prevent the person-to-person spread of COVID-19. This identifies and notifies persons with a confirmed diagnosis and their primary contacts.

Uganda has, over the years, developed contact tracing strategies for Ebola, Marburg and Crimean-Congo Haemorrhagic fever outbreaks.[8] With such strategies, the surveillance teams are able to identify and isolate cases, and their primary and secondary contacts. However, this 'shoestring epidemiology' is too slow for fast-evolving epidemics and pandemics such as COVID-19.

To improve the speed and efficiency of conventional contact tracing, geolocation features integrated within mobile phone have been used to track proximity which can translate into contact once one becomes a case for a disease of interest.[9] This digital contact tracing (DCT) approach automatically notifies individuals who test positive and their immediate contacts to test and/or isolate. It is worth noting that population-wide uptake

of such technology is incumbent on the level of smartphone ownership, possibly the current bottleneck for low-income countries like Uganda. Therefore targeted use becomes the most viable alternative, however, such technology is inherently public-facing as it harnesses information directly from mobile phones.[10 11] This inevitably comes with ethical, legal and sociological issues that must be robustly addressed to ensure sustainable adoption for public health. In a nutshell, DCT deployment requires a delicate balance between public health benefit, which is fairly distributed against the risk of harm and basic principles of ethics.[12 13]

In countries like the USA, various forms of this technology have been used to support contact tracing with varying levels of success. Although success depends on the scale of deployment, better outcomes have been reported with targeted use for private companies, particularly in Silicon Valley and sports franchises like American football.[14 15] Therefore, countries intending to adopt this technology ought to examine and respond to the following evaluation questions; (a) Can a DCT tool be effective in augmenting the current public health responses? (b) If so, to what degree? (c) Which specific aspects? (d) With what confidence? and (e) What are the prerequisites that ensure a DCT tool delivers on public health goals in a way that is ethically, legally and socially defensible?.[16]

Here we target the haulage sector with the aim of showing how DCT can be developed and deployed to support public health needs in a way that meets the recommended sociological, ethical and legal boundaries in East Africa.[17] We have started by developing a DCT tool prototype, with input from the relevant stakeholders; this will be further developed and tested in Uganda (Link to download the current prototype; (https://www.project-thea.org). It is noteworthy that our feedback will be collected through stakeholder meetings to be held throughout the project lifecycle (figure 2).

Our team will also examine the interaction between end users' (knowledge, attitudes and practices, KAP) and societal context (bioethics and legal) regarding COVID-19 and how this influences technology uptake and ultimately impacts COVID-19 mitigation and infection control measures. We target the haulage sector based on the hypothesis that our DCT tool can harness their unique but predictable mobility using mobile phone technology embedded with: (a) road network infrastructure, (b) time-stamped geopositioning and (c) COVID-19 test result data. Together, these parameters will be optimised to increase the speed and accuracy of public health contact tracing in this region.

This will be achieved through the following specific objectives:

1. To conduct stakeholder consultative meetings that inform the design of a haulage sector tailored-DCT.
2. To develop the DCT technology.
3. To test the utility of the DCT among truck drivers.

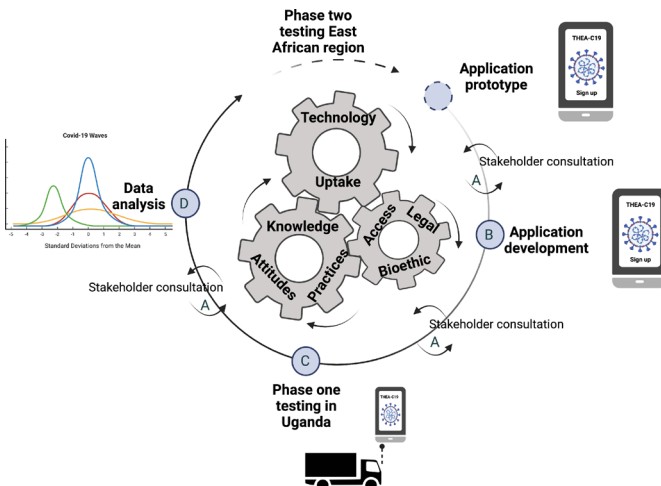

**Figure 2** The conceptual framework on which this study is developed and will be implemented. The three interlocking wheels are used metaphorically to represent interlinkages between technology, individuals and communities. We posit that technology update is enhanced by understanding inherent characteristics of the intended user (attitudes, knowledge and how these translate into actions) and the characteristics of the community they inhabit (legal, ethical, traditions and connectivity access). The project has four components: (A) stakeholder engagement and inclusion, (B) technology development, (C) testing the technology and (D) data analysis to support public health interventions. The image was generated using Biorender https://biorender.com

4. To develop models for mapping risk, transmission and estimate haulage attribution to COVID-19 epidemiology in Uganda.
5. To conduct stakeholder dialogue with feedback and review of potential benefits and harms of DCT technology.

## METHODS
### Study design
The COVID-19 pandemic presents a continuously changing epidemiological landscape; therefore, the study designs is adaptable to accommodate unpredictable circumstances. The project will generally use a mixture of methods tailored to each of the project components. For example, (i) stakeholder engagement will use qualitative and quantitative approaches to generate data. This will be done on a representative number along the hierarchy of the stakeholders in a cross-sectional study, (ii) develop a DCT tool with stakeholder feedback for contextualisation and (iii) a longitudinal study to test the DCT tool with truck drivers will capture data using the mobile application.

### Study setting
The field study will start in January 2022 for 1 year, primarily focusing on the East African haulage network, particularly that which transits through Uganda (figure 3A). Uganda has 52 POEs, with varying levels of haulage traffic, categorised as (i) high, (ii) moderate and

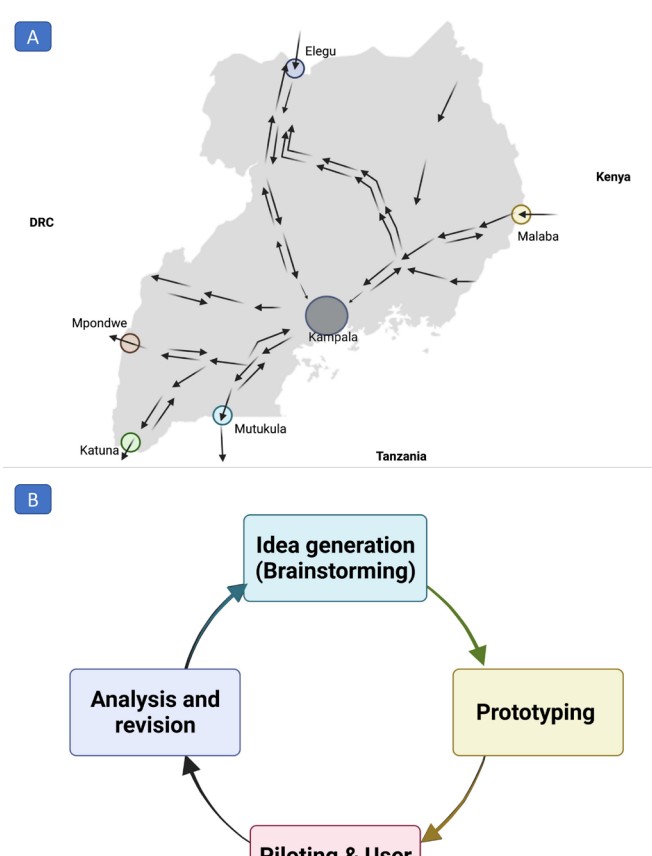

**Figure 3** (A) The convergence of the East African haulage network in Uganda. The study will focus on the six ports of entry and exits (POEs) shown as circular highlights. (B) The iteration stages of the human-centred design approach. The image was generated using Bio render https://biorender.com.

(iii) low-traffic POEs. This study will focus on one seclusion point in Kampala and five POEs, with the heaviest traffic routes with a combined haulage network traffic coverage of 45%.

### Study participants
*Identification and recruitment of key stakeholders*: The key are local and regional haulage truck drivers. Other stakeholders include public health experts, including port-health and surveillance departments of the Ministry of Health, the National Logistics Platform (NLP), Ministry of Transport, border point entry clearance and exit officials, legal, internet security, ad software developers and bioethicists as well as local leaders within neighbouring communities of service stations. There are approximately 123 000 registered haulage truck drivers in Uganda, 15% of whom are involved in regional haulage.[18] Study participants will be recruited with guidance from the NLP.

### Sample size
*Formative research study*: We will recruit a diverse sample of truck drivers from any of the six study sites (figure 3A). Other stakeholders will be purposively selected to ensure diverse experience and expertise. We estimate that a

maximum of 30 participants will contribute to three focus group discussions (FGDs), with another 6–12 individuals taking part in the key informant interviews (KII), while the self-administered questionnaire-based survey will generate data from approximately 300 respondents. The sample size estimation for examining knowledge attitudes and practices related to COVID-19 among haulage drivers and allied stakeholders will be done based on the methodology published by this group.[16]

*Quantitative study*: We shall conduct a survey regarding stakeholder input on the mobile application's utility, functionality and privacy. We will aim to recruit about 65% of the ~123 000 truck drivers to test the utility of the mobile application. This estimate is based on data from counties where this type of technology has been successfully tested on approximately 60%–80% of the target population .[18–20] If we target 65% of the study population, this will represent 11 200 (regional haulage drivers) and 72 000 (local haulage truck drivers), respectively.[21] Our inclusion criteria will be (a) consent to participate in the study and (b) ownership of a smartphone that runs on Android or MAC IOS operating systems.

## Procedures

*First objective aims to conduct stakeholder consultative meetings to inform the design of the DCT tool tailored to the haulage sector in Uganda.* To develop technology that embodies users' values, we shall conduct three stakeholder consultative meetings over the next 18 months. Using a stratified sampling method, we will identify a representative sample of key stakeholders such as regional and local haulage truck drivers, conductors and border point entry and exit clearance officials with varied characteristics that might modify the effects of the tool. These will include the individuals' age, experience, gender and education (figure 2). It is noteworthy that we shall limit face-to-face interactions and use digital communication as much as possible, including hosting virtual structured discussions at regional and local entry and exit ports; and project meetings between partners.

## Patient and public involvement

In addition to engaging key stakeholders as described above, we will engage research participants in developing and testing a mobile-based DCT tool that incorporates their input for contextualisation.

*The first workshop* will focus on: (a) mapping the KAP around COVID-19 among stakeholders in the haulage industry. This will be done using KII, structured questionnaires, as well as break-away FGD. (b) In addition, we shall assemble a panel of communication experts to develop awareness and sensitisation materials targeting the study population. We will focus on empowering participants with the knowledge needed to protect their ethical, social and legal boundaries with regard to technology. (c) We shall also conduct a survey to collect stakeholder input on the functionality and privacy settings of our mobile application.

*The second workshop* will focus on informed consent, such as how one voluntarily joins or withdraws from the study, optimise parameters suitable to protect personal freedoms, and highlight in-built ethical and technology boundaries. It is noteworthy that the KAP survey will be done at all consultative meetings. The second FGD comprises long-distance truck drivers. In contrast, the third comprises public health experts to maximise feedback on the functionality and utility of the mobile application. Approximately, 30 participants will account for saturation with regard to FGDs,[22] although each meeting will include only 10 participants at a time, lasting not more than an hour.

*Second objective will be to develop a DCT tool to support COVID-19 surveillance.* This process uses feedback from *objective one*; to inform the development of the DCT tool. We will employ a human-centred design approach— an iterative process of idea generation, prototyping, user-testing and piloting that involves end-users in the development process[23] (figure 3B).[24] The application will be developed to embody values that maximise user anonymity, flexibility and minimal data storage and battery life.

*Third objective will be to test the utility of the DCT tool among haulage truck drivers.* This will be achieved through truck drivers' interaction with the tool. Once the mobile application is downloaded and installed by the driver, it will (a) ask for consent from the driver to access the global positioning system (GPS) and (b) request for information on regular haulage routes such as departure and destination.

In this regard, the mobile app will have a feature that a user can choose to allow its activation whenever movement relative to the road network is detected that is, 'auto-activation when on road network'. Finally, (c) the application will then generate a universally unique identifier (UUID) or 'digital token' as a pseudonym and deposit it into the secure, centralised database. Once processes (1) to (2) as shown in (figures 4 and 5) are completed, the app will launch and run in the background.

As an option, the app will also request the user for their metadata (age, sex, district of residence and phone contact), which will also be stored encrypted on the phone. The three pieces of information that is, UUID, sampled GPS and metadata are kept encrypted separately. It is noteworthy that the regional haulage truck drivers will be required to have a negative test valid for 14 days, although a driver can cross the border 2–4 times a month.[17]

*COVID-19 testing*: COVID-19 testing and reporting at POEs, as shown by steps 3–5 in figure 5, will be conducted by public health officials. Here we will collaborate with all Ministry of Health approved laboratories for testing truck drivers at POEs in Uganda, and these include (a) test and fly and (b) Maia Medical laboratories in Kampala, Mpondwe, Mutukula, Malaba and Elegu. Before a sample is collected for the PCR test from the driver, consent will be sought and then the THEA-C19 App will be installed on the smartphone. The App scans a QR-code of the East

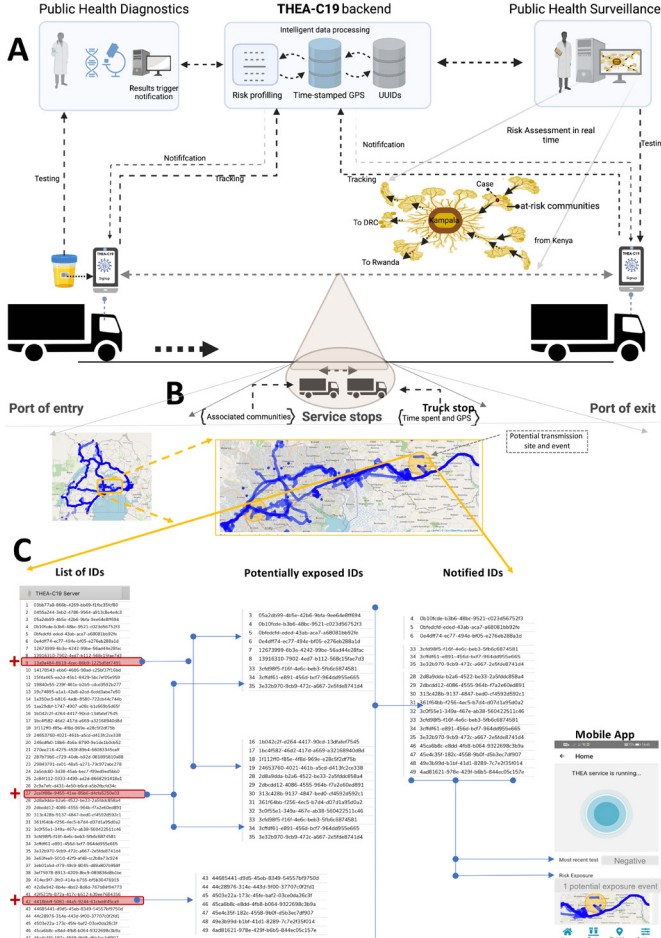

**Figure 4** (A) The architecture of the digital contact tracing tool how stakeholders feed or receive data on the system. (B) An example of the generated contact graph from the time-stamped GPS locations relative to the road infrastructure. In yellow, it highlights the hotspot where contact with an individual who tested positive occurred. (C) The risk assessment process based on a decision tree; this identifies exposed individuals that the systems notify. UUIDs, universally unique identifier.

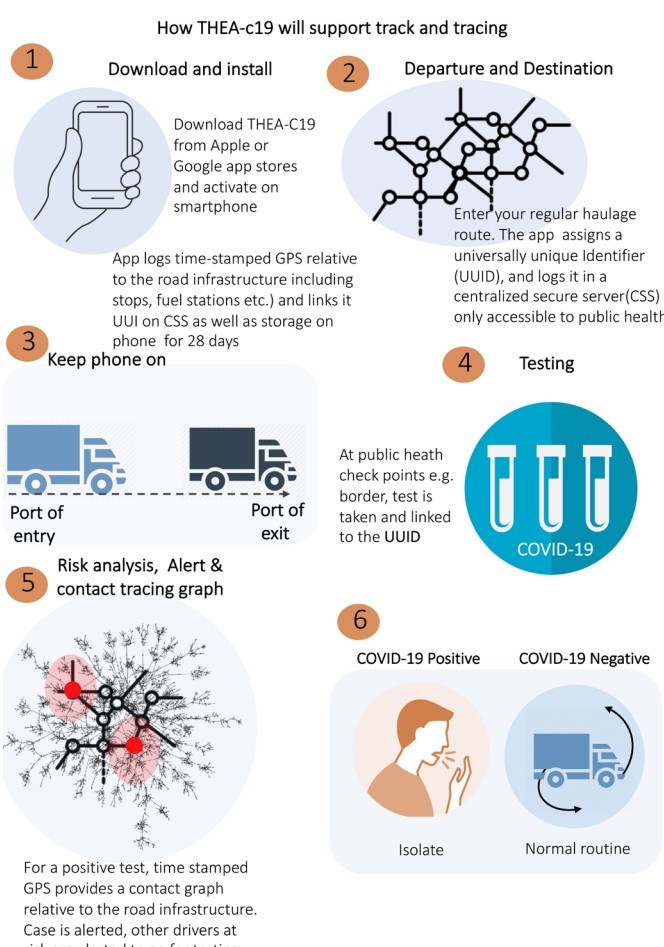

**Figure 5** The THEA-C19 mobile application installation and contact tracing processes step by step.

Africa truck driver's number used as laboratory sample ID. The laboratory will then post the results into the centralised results dispatch system (RDS) managed by the Ministry of Health. The scanned QR-codes will generate a list of ID's, which our system will use to retrieve results from the RDS. Here, the DCT system will do the following:

1. Retrieve and notify the driver of their results.
2. If the result is positive, the driver will receive a notification and the current guidance by the ministry of health as a reminder.
3. Public health will then use the tracking data to create a 3-day risk map for cases which will be available as a web interface accessible to the public health surveillance team to support decision making.
4. Once notified, the case and contacts are required to quarantine based on the guideline, adherence will be assessed using the inherent geo-fencing on road network. For example, if a notified case or their contact

are detected on the road network within 10 days from notification date, our system will flag this as an event of non-adherence subject to further investigation by a public health official.

When fully functional, the system can generate time logs for events and their notifications, and this data allows us to evaluate the overall speed of the DCT process. We can also identify time lags between critical steps: for example, on average, it takes 1 hour for a test result to be posted in the result dispatch system (RDS) of the ministry of health. The tool can then calculate the time from result posting to case notification, risk assessment and contact notification. For technical reasons, the THEA-server will query the RDS every 30 min to retrieve new results; this means result notification could take up to 1 hour and a half from sample collection and contact notification up to 1 hour after that. The frequency of notification will be optimised for prudent computing capacity allocation for risk assessment.

Notably, the output can inform future risk-based screening, which would save haulage companies money and time in testing.

*Contact tracing:* This will be implemented via the web interface only accessible to the Ministry of Health, figure 4A. The interface will generate a contact graph

from the time-stamped GPS locations relative to the road infrastructure as shown in figure 4B and step 5 (figure 5) and overlay it with the COVID-19 test results to reveal the risk trail cases in the last 3 days. For example, three people tested positive, as shown in figure 4C. The system will generate a list of all drivers, including the positive in the previous 3 days on the road network. Using a decision tree approach, checks where they stopped and for how long and which stops overlap with the positive drivers. This will then generate a list of potentially exposed individuals which will then be ranked by the time of overlap. Drivers with the most extended time overlap at a given location will then be notified via the mobile application, considered as most likely exposed (figure 4C).

*Community risk assessment:* Our system will generate a list of potentially exposed drivers for every positive test. These potentially exposed drivers represent the tool's prediction which in reality should be linked to a transmission event. The system will then compute the number of potentially exposed drivers on the list who test positive within the following 7–14 days to generate a 'percentage predicted transmission (PPT)' metric. Since a transmission event is associated with a location linked to communities on the road network, over time, public health can use the PPT to predict/estimate the risk for the communities liked to the road network. Notably, we estimate that asymptomatic and presymptomatic cases will account for 15% and 52% of the notified positive cases, respectively,[25] which are associated with a lower secondary transmission risk. Such estimates will be contextualised using our empirical data and used to evaluate the tool's effectiveness for contact tracing, that is, the sensitivity and proportion of expected false positive contact notification.

*Fourth objective will be to develop models for mapping risk, infer transmission and estimate a haulage attribution:* The data collected by the mobile application (DCT) will be analysed as follows: first, we will use machine learning methods to compute the probability of drivers testing positive as a function of characteristics of their movements (eg, their route, duration of the journey, the departure time, vehicle type (~fuelling frequency) estimated traffic and contact structure). Determining a probabilistic output will enable the public health officials to prioritise who to test (risk-based diagnostics), limiting the number of contacts between infected drivers and individuals of the community. Note that since this process will need a training data set, the first 4 months will be to generate this data set (~160 000 trips). Therefore, the utility of this approach for this analysis will be after 6 months of data collection.

Second, the movements of trucks along the road networks will form a large network where nodes such as villages/towns/cities and edges are movements of haulage vehicles. This network will then be analysed to evaluate which communities are key in the structure of the network, their potential for disease transmission and infection as well as their potential as candidates for placing efficient testing facility outposts.

Finally, assuming all drivers test negative at departure, we will estimate the likelihood of infection from communities along each individual travel route based on the spatial and temporal aspects of each individual journey, the time of testing, factoring in the epidemiological characteristics of COVID-19 (latent, incubation and symptomatic periods) in addition to probability of clinical signs. Based on this evidence, communities with a high risk of infection (either as source or recipient of infection) will be identified. Such a risk assessment would provide timely evidence for targeting communities with high risk of disease and improve resource allocation. By combining this with other data, we shall estimate the contribution of haulage to the epidemic and evaluate the impact of using this intervention on COVID-19 control, however the last component of the analysis will likely occur beyond the 18 months project period.

*Fifth objective is to conduct stakeholder dialogues to explore the impact of the mobile application (DCT) on disease control and preparedness.* Deliberative dialogues are a form of public discourse that are currently used to develop a shared understanding of a problem or solution among key stakeholders.[26] The objectives of these dialogues will 'not so much to talk together as to think together, not so much to reach a conclusion as to discover where a conclusion might lie' and to ensure that different viewpoints and assumptions are explored.[27] Dialogues also provide opportunities for tacit knowledge to be shared.[28 29]Using deliberative dialogues, we will create spaces where study findings are discussed and explored by key stakeholders, provide opportunities to examine the extent to which the study findings are applicable to their contexts and how use of the mobile application technology could be formalised and scaled up to ensure appropriate governance and mechanisms to maintain data privacy. These stakeholder dialogues will follow the same approach as described for the KAP above.

*Formative research data analysis* will be done using interpretative description approaches,[30] such as those used in ethnography and naturalistic inquiry to analyse data arising from the KAP survey, KII and FGDs targeting the stakeholders. These approaches allow us to examine the practical implications of DCT on the routine activities of the truck drivers in the context of the existing body of knowledge in an iterative analysis, while exploring challenges and opportunities to using DCT technologies. We shall use qualitative analysis software, such as Nvivo to code and interrogate our data.

## DATA COLLECTION, PROTECTION AND MANAGEMENT
The team will take the following steps to secure the mobile app and THEA-C19 server:
1. Use transport layer security for all communication between the mobile app and THEA-C19 server.
2. Add rate limiting to the application program interface(API) service to prevent denial of service attacks and ensure availability at all times.

3. Maintain an access log for all system accesses.
4. Sanitise and validate all data submitted to the server to prevent cross-site scripting and SQL injection attacks.
5. Use the latest version of the frameworks, runtimes and database software to ensure the latest security patches are in place. Actively monitor common vulnerabilities and exposures(CVE) disclosures for all software packages used in order to respond to any reported vulnerabilities quickly.

There are extra security layers embedded which are not disclosed here to ensure that the utility of the tools is tamperproof.

The contact tracing component will primarily use the collected time-stamped GPS data, which will be strictly limited to the road network, and are securely collected and transmitted to a server. The data collected will be stored in a relational database and will be retrievable in a number of formats including (comma separated values)CSV for analysis. The database will be replicated online to provide redundancy and ensure uninterrupted operation. In addition, daily backups will be taken in the event that we need to restore the data which will be kept for the duration of the project. Use of the data will only be triggered by a case; it is this case's risk profile that informs the contact tracing and notification via the mobile application. Otherwise, all data will be kept separate. Diagnostic data will be generated and posted to a secure public health centralised system also referred to as the RDS.

The stakeholder engagement component will generate qualitative data, collected as part of the KAP analysis linked to COVID-19, ethics and legal aspects of technology deployment. It will consist of audiorecordings, observational notes, transcribed texts and still photography. Participants will be required to consent before downloading or installing the app. All participant data will be anonymised and stored securely on our servers for analysis. In addition, sequence data will be generated from a selected number of samples using Oxford Nano pore; this data once superimposed on the network allow us to monitor the routes of COVID-19 strain introduction. The processing of this data will generate secondary data to be securely stored at the African Center for Excellence in Bio Informatics at the Infectious Diseases Institute at Makerere University.

## PROJECT DELIVERABLES
### Open source DCT tool
We shall develop an open source DCT tool tailored to haulage in Uganda to enhance the speed and efficiency of the current manual of public health contact tracing system. In so doing, we shall redirect resources from manual tracing, subsequently reducing border transit time for haulage while indirectly supporting increased supply chain flow.

### Statistical and mathematical models
Furthermore, the modelling team will develop statistical and mathematical models using output from testing for epidemic prediction and risk mapping. The outputs will provide evidence to inform (a) public health decision-making on personnel and resource allocation critical to improving the efficacy, cost-efficiency and ultimately safety and surveillance of contact tracing activities, (b) future control strategies to mitigate the spread of COVID-19 and other subsequent novel pathogens while limiting economic consequences and strengthening the resilience of the commodities' supply chain.

### Avail evidence for vaccination programmes
The contact tracing analysis will provide epidemic characteristics, such as the speed of spatial transmission and size of the epidemic in specific locations, for example current data on hotspots. The data can then be used to define prioritisation of vaccination, estimate vaccination coverage while providing evidence on vaccination impact (ie, temporal and spatial changes) in the epidemic size.

### Avail evidence for policy
Finally, any lessons learnt regarding the legal, ethical and social–anthropological examinations related to DCT deployment will be used to regulate usage and inform policy formulation in relation to use of DCT technology in combatting pandemics.

### Interpretation and reporting of findings
Building on the deliberative dialogues, a small group of key stakeholders will be engaged in interpreting and confirming interpretations of findings. They will be invited to comment on reports and be coauthors.

The development of our digital contact traing tool is complete, and we are currently conducting a field study to test the functionality. In addition, our teams are currently conducting the ethical, social and legal dimensions of deploying such a tool for public health in the Kampala district. The project is on track to deliver on the aspects listed above.

## ETHICS AND DISSEMINATION
In Uganda, ethics approval was obtained from School of Public Health Higher Degrees, Research and Ethics Committee at Makerere University (approval number SPH-2021-35) and Uganda National Council for Science and Technology (approval number HS156ES). At the University of Edinburgh, study approval was granted by the Human Ethics and Research Committee at the Easter Bush Campus (HERC-538).

We will obtain informed consent for participation in interviews, FGDs, still photography and videography. Data from interviews, questionnaires, GPS, photos, videos and notes will be anonymised.

### Benefits of participation
Since clinically valid tests such as COVID-19 PCR test are part of this protocol, research participants may benefit

from individual results provided conveniently without financial cost. However, since research participants will not be provided any therapeutic interventions, consent forms clearly state that no financial benefits shall be gained as a result of individual participation in this research study, however haulage companies may ultimately benefit monetarily by reducing their employees' exposure to the virus thereby limiting sickness-related absences while maintaining operations.

## Risks of participation

In general, risks of participation resulting from both psychological and physical intervention necessary to acquire a biological sample are trivial. The primary risks to participation in this study are generally psychological, such as stigma and discrimination if sensitive information is generated in the project and there is a breach of privacy or confidentiality. However, industry standards that ensure confidentiality and privacy will be observed and maintained at all levels of data collection and processing to further mitigate that risk. No personal identification details will be linked to individuals as participants will be assigned unique identification pseudonyms, 'digital tokens'. The project will be conducted in accordance with the Ministry of Health and Uganda National Council for Science and Technology guidelines for the prevention of spread of COVID-19.

We will provide all participating organisations with a report of the main findings as soon as the analyses have been completed and independently verified by at least two referees who are not part of the study team.

We will actively disseminate the results of this study through publications and presentations. At National level, the study is nested within the Ministry of Heath, with study outputs directly feeding into the public health responses via incident command teams. At the regional level, study findings will be presented at regional haulage sector meetings facilitated by the NLP in Uganda. Internationally, we will actively disseminate the study findings through peer-reviewed publications, pre-existing partnerships and our organisation websites such as https://project-thea.org/ and Github account for the open source code https://github.com/project-thea/.

**Author affiliations**
¹The Roslin Institute, The University of Edinburgh The Roslin Institute, Roslin, UK
²Blockchain Technology Laboratory, The University of Edinburgh School of Informatics, Edinburgh, UK
³Makerere University College of Humanities and Social Sciences, Kampala, Kampala, Uganda
⁴Department of Medicine, Makerere University College of Health Sciences, Kampala, Uganda
⁵Institute of Health and Society, Faculty of Medicine, Universitetet i Oslo, Oslo, Norway
⁶National Health Laboratories and Diagnostic Services, Antimicrobial Resistance National Coordination Centre (AMR-NCC), Ministry of Health, Kampala, Uganda
⁷The University of Edinburgh The Roslin Institute, Edinburgh, UK
⁸VetAgro Sup, Marcy-I'Etoile, Auvergne-Rhône-Alpes, France
⁹Digital infrastructure & Software development, Bodastage Solution, Kampala, Uganda
¹⁰School of Medicine, Makerere University College of Health Sciences, Kampala, Uganda
¹¹Immunology and Molecular Biology, Makerere University College of Health Sciences, Kampala, Uganda

**Contributors** Conceptualisation: AM, ERS, MJ and IM. Funding acquisition: AM, CM, IM, BMdB, TP, ERS, AK, ESM and MJ. Investigation: AM, CM, AN, IM, BMdB, TP, ERS, AK, ESM and MJ. Writing-original draft: AM, CM, AN, IM, BMdB, TP, ERS, AK, ESM and MJ. Writing-review and editing: AM, CM, AN, IM, MB, TP, ERS, AK, ESM and MJ.

**Funding** This study is funded by the National Institute of Health Research/Medical Research Council-GECO award, ref MR/V034952/1.

**Map disclaimer** The inclusion of any map (including the depiction of any boundaries therein), or of any geographic or locational reference, does not imply the expression of any opinion whatsoever on the part of BMJ concerning the legal status of any country, territory, jurisdiction or area or of its authorities. Any such expression remains solely that of the relevant source and is not endorsed by BMJ. Maps are provided without any warranty of any kind, either express or implied.

**Competing interests** None declared.

**Patient and public involvement** Patients and/or the public were involved in the design, or conduct, or reporting or dissemination plans of this research. Refer to the Methods section for further details.

**Patient consent for publication** Not required.

**Provenance and peer review** Not commissioned; externally peer reviewed.

**ORCID iDs**
Adrian Muwonge http://orcid.org/0000-0002-8579-0384
Allen Nsangi http://orcid.org/0000-0001-8702-9217

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
