## [Reviewer comments · BMJ Open]

ARTICLE DETAILS

TITLE (PROVISIONAL)	A protocol for developing digital contact tracing tailored to haulage in East Africa to support COVID-19 surveillance
AUTHORS	Muwonge, Adrian; Mpyangu, Christine; Nsangi, Allen; Mugerwa, Ibrahim; Bronsvoot, Barend M. deC.; Porphyre, Thibaud; Ssebagala, Emmanuel; Kiayias, Aggelos; Mwaka, Erisa; Joloba, Moses

VERSION 1 – REVIEW

REVIEWER	Kilmarx, Peter John E Fogarty International Center
REVIEW RETURNED	19-Dec-2021

GENERAL COMMENTS	The authors are to be commended for planning a study of digital contact tracing among haulage drivers in Uganda. They propose to conduct formative studies and disseminate and evaluate a mobile phone GPS digital contact tracing solution in a substantial fraction (65%) of the 123,000 truck drivers in Uganda. Comments: 1. Line 58-59: Is the high proportion of COVID-19 from among the haulage network because of more intensive testing in that group, i.e., ascertainment bias?2. The paper mentions testing every 14 days on Line 187 and 2-4 times a month on line 375. Which is it? Testing once every 14 days seems too infrequent. Line 528-529 states that the study will provide the testing. More details are needed. Where will specimens be collected? What manufacturer's(s') test(s) will be used?3. It's unclear how digital contact tracing would alleviate the bottlenecks at the ports of entry. How would contact tracing obviate the need for a negative test before entry?4. Line 296-303: Would call this "formative research." It is not only qualitative research because it includes self-administered questionnaires which presumably have some quantitative elements.5. Line 391: More detail is needed on the specifications of the proposed digital contact tracing intervention. What are the proposed parameters to define a contact? Proximity of less than two meters for greater than 15 minutes? Presence in the same general location on the same day?6. Why develop a new digital contact tracing application? Why not use existing widely available apps such as the Google Apple Exposure Notification (GAEN)-based apps or GPS apps already developed?7. A logic model and some preliminary modeling are needed to estimate the potential impact of the proposed intervention. The
--

	examples given, tech companies and sports franchises, are relatively closed systems where contacts are largely within group. It's unclear how digital contact tracing would impact a large, widely dispersed group with extensive out-of-group contacts. What proportion of cases will be detected during pre-symptomatic or asymptomatic infectious periods? What is the usual turnaround time for test results? What is the expected time lag for contact notification? What will contacts be expected to do? Quarantine in place? Test more frequently? 8. The expected outcomes should be described. How many positive tests are anticipated during high and low transmission periods? How many contacts would be identified? What are the targets to define success? 9. Will the response of contacts be monitored? They could be called or sent questions by text messages to see if they get tested or are quarantined. Could their movement be monitored by GPS? 10. Line 500: It's unclear how the data on highly mobile drivers with weekly or fortnightly testing will provide information on local geographic transmission. 11. Figure 2: Three interlocking gears won't turn.
--	---

VERSION 1 – AUTHOR RESPONSE

Reviewer: 1

Dr. Peter Kilmarx, John E Fogarty International Center

Comments to the Author:

The authors are to be commended for planning a study of digital contact tracing among haulage drivers in Uganda. They propose to conduct formative studies and disseminate and evaluate a mobile phone GPS digital contact tracing solution in a substantial fraction (65%) of the 123,000 truck drivers in Uganda.

Comments:

1. Line 58-59: Is the high proportion of COVID-19 from among the haulage network because of more intensive testing in that group, i.e., ascertainment bias?

Response. We acknowledge that this is a possibility; however, Uganda adopted a risk-based sampling (targeting truck drivers) because of their mobility and case incidence. Given that this is a landlocked country, the source of disease/variants was viewed as external. So mobile groups (travellers and haulage) were more likely to introduce infection and variants, hence why they are targeted. This risk-based sampling strategy has been maintained as the countries control new strain introductions. Regardless of community infection rates in Uganda, truck drivers would play a critical role in introducing infection from neighbouring Congo and South Sudan, with fragile public health systems. However, the issue raised by the reviewer will be important during the analysis and interpretation/generalisation of our results.

2. The paper mentions testing every 14 days on Line 187 and 2-4 times a month on line 375. Which is it? Testing once every 14 days seems too infrequent. Line 528-529 states that the study will provide the testing. More details are needed. Where will specimens be collected? What manufacturer's(s) test(s) will be used?

Response. The drivers will test twice a month, and this is because each negative test is valid for 14 days. We acknowledge that a driver may cross the borders 2-4 times a month, which means some results will be used for more than one border crossing as long as the crossing is less than 14 days. This has been clarified on line 359

3. It's unclear how digital contact tracing would alleviate the bottlenecks at the ports of entry. How would contact tracing obviate the need for a negative test before entry?

Response: The truck drivers queued up because they needed to wait for their results at the border where they tested, which would take 24-48 hours. The new approach(as present in lines 181-185) allowed them to continue with the journey and send the result electronically. There had to be fast contact tracing if one tests positive for this to happen. So, digital contact tracing was the measure proposed to ensure that risk/impact due to such cases is limited.

4. Line 296-303: Would call this "formative research." It is not only qualitative research because it includes self-administered questionnaires which presumably have some quantitative elements.

Response: We thank the reviewer for this correction, this nomenclature has now changed please see line 283 & 444

5. Line 391: More detail is needed on the specifications of the proposed digital contact tracing intervention. What are the proposed parameters to define a contact? Proximity of less than two meters for greater than 15 minutes? Presence in the same general location on the same day?

Response. We have now provided details on the functionality and illustration of the digital contact tracing tool see **Fig.4** and text on line 381-392 & 393-399

6. Why develop a new digital contact tracing application? Why not use existing widely available apps such as the Google Apple Exposure Notification (GAEN)-based apps or GPS apps already developed?

Response

The novelty of this work is developing technology with the intended user. It allows us to explore the complexities of technology adoptability navigating technology's ethics, legal and socio-cultural dimensions. This would probably be the first time such technology has been developed for public health using this approach on the African continent. From a technological point of view, while 80% of people in East Africa have access to mobile phones, only about 45% will be smartphones. It is noteworthy that this number is rapidly rising as a response to the local mobile phone innovations(such as mobile money payments). The haulage sector is one of the sectors with fast-growing technological

innovation; therefore, truck drivers are rapidly adopting smartphone use. Importantly, our technology emphasizes anonymity, simplicity and flexibility developed with user input. It uses time-stamped GPS, road-network restricted tracking, and test results to provide digital contact tracing and road risk assessment. We believe this provides unique utility for such a target group and inform the safe deployment of digital contact tracing technologies needed for combatting pandemics in low-income countries.

7. A logic model and some preliminary modeling are needed to estimate the potential impact of the proposed intervention. The examples given, tech companies and sports franchises, are relatively closed systems where contacts are largely within group. It's unclear how digital contact tracing would impact a large, widely dispersed group with extensive out-of-group contacts. What proportion of cases will be detected during pre-symptomatic or asymptomatic infectious periods? What is the usual turnaround time for test results? What is the expected time lag for contact notification? What will contacts be expected to do? Quarantine in place? Test more frequently?

Response: This is an excellent question; while truck drivers are not part of a closed bubble, they are part of an epidemiological carousel through Uganda's road network. We target the haulage sector based on the hypothesis that our mobile phone digital contact tracing tool can harness their unique but predictable mobility patterns. This would then help establish their role in introducing, maintaining, and introducing infection and new strains. It is noteworthy that the tool is primarily developed for contact tracing. Its utility is best harnessed during low disease incidences in the communities. We have provided preliminary output from this tool, as shown in **Fig 4C**. This shows the output as seen by Public health Surveillance and the result on the app. It is worth noting that some of the questions asked are answered as the tool is being tested as part of this implementation research.

In addition to the primary utility, we anticipate that the tool can act as an early warning system by tracking Percentage predicted transmission(PPT) as computed by the tool. This parameter can be used also to validate how well the tool is performing with regards to assigning risk of exposure as described in lines 393-399

8. The expected outcomes should be described. How many positive tests are anticipated during high and low transmission periods? How many contacts would be identified? What are the targets to define success?

Response: The outcomes were initially challenging to forecast; however, very early data suggests that over 90% of the drivers that cross the POEs take a test and results are posted in the central database (RDS). During this Omicron wave, from January 2022, when we started registering drivers on the DCT system, 1627 drivers have been registered in the system. Of these, 50 have tested positive. The figure below shows our public health surveillance dashboard; please note that some drivers will be registered when they have a valid travel result, so a test will not be taken on that day. The second panel shows a filter of tests to show positives (~ 3% positivity rate in this group). Please note that the cases per 1M at a national level is 0.89 as of 21 February 2021 <https://ourworldindata.org/coronavirus/country/uganda>

The example show in figure 4C, out of 49 drivers on the road on that day, 18 were notified. To define success, we shall use the PPT, i.e. of the 18, the number of drivers who test positive within 7-10 days of notification. However, this component is still being refined.

9. Will the response of contacts be monitored? They could be called or sent questions by text messages to see if they get tested or are quarantined. Could their movement be monitored by GPS?

Response- The tools notify potential contacts, as shown in Fig 4. They are monitored just like any other driver on the system; only when they test positive do we notify them to follow the guidelines stipulated by the Ministry of health. The message notification is currently restricted to results as shown in Fig 4C, this can be modified to include other notifications if the feedback from the primary users highlights it.

10. Line 500: It's unclear how the data on highly mobile drivers with weekly or fortnightly testing will provide information on local geographic transmission.

Response- As shown in Fig 4C, the system generates a list of potentially exposed drivers; these potentially exposed form what the tool predicts to represent a transmission event. If individuals on that list test positive within the next 7-10 days, the system can compute "a percentage predicted transmission" the number of predicted exposures that test positive. Over time, we can establish how this figure changes and because these transmission events are likely to occur on the road infrastructure. Fig 4C represents risk to associated communities. So the predicted transmission parameter becomes a metric for the level of risk to the communities.

11. Figure 2: Three interlocking gears won't turn.

Response. Physics & Mechanics 101, well spotted. Indeed, three interlocking gears produce limited movement. On this occasion, we are using this metaphorically, three parts that contribute to a unified motion, recognising that some parts, if not well attended to, can grind the whole approach to a halt.

Reviewer: 1
Competing interests of Reviewer: None.